# STAT3 is a genetic modifier of TGF-beta induced EMT in KRAS mutant pancreatic cancer

Stephen D'Amico, Varvara Kirillov, Oleksi Petrenko, Nancy C Reich*

Department of Microbiology and Immunology, Stony Brook University, Stony Brook, United States

**Abstract** Oncogenic mutations in KRAS are among the most common in cancer. Classical models suggest that loss of epithelial characteristics and the acquisition of mesenchymal traits are associated with cancer aggressiveness and therapy resistance. However, the mechanistic link between these phenotypes and mutant KRAS biology remains to be established. Here, we identify STAT3 as a genetic modifier of TGF-beta-induced epithelial to mesenchymal transition. Gene expression profiling of pancreatic cancer cells identifies more than 200 genes commonly regulated by STAT3 and oncogenic KRAS. Functional classification of the STAT3-responsive program reveals its major role in tumor maintenance and epithelial homeostasis. The signatures of STAT3-activated cell states can be projected onto human KRAS mutant tumors, suggesting that they faithfully reflect characteristics of human disease. These observations have implications for therapeutic intervention and tumor aggressiveness.

*For correspondence:
nancy.reich@stonybrook.edu

Competing interest: The authors declare that no competing interests exist.

## eLife assessment

This study delves into the complex role of STAT3 signaling and its interplay with TGF-beta and SMAD4 in KRAS mutant pancreatic cancer. The authors demonstrate that both the presence and absence of STAT3, relative to SMAD4, can lead to poor PDAC differentiation and that STAT3 mutations affect p53-null fibroblasts with KRASG12D and induce an EMT-like phenotype. By providing **convincing** evidence, the authors were able to derive **important** insights into KRAS mutant cancers.

## Introduction

Pan-cancer projects, such as The Cancer Genome Atlas (TCGA), have provided a comprehensive view of the mutational landscape in human cancers. The foremost objective has been the discovery of key genes that drive cancer initiation and progression. It is estimated that cancer genomes contain an average of less than five driver mutations, whose outcomes are realized in the context of chromosomal and epigenetic alterations (*Sondka et al., 2018*; *Bailey et al., 2018*; *Consortium ITP-CAoWG, 2020*). While the Cancer Gene Census (CGC) has been largely defined, unraveling the contributions of normal cell functions to cancer development and their influence on tissue homeostasis, plasticity, and differentiation remains a complex task. A large body of evidence suggests that signal transducer and activator of transcription 3 (STAT3) has tumor-promoting properties that it exerts in a context-dependent fashion (*Yu et al., 2014*; *Huynh et al., 2019*). Canonical activation of STAT3 occurs following phosphorylation of tyrosine 705 (pY705) by receptor-associated Janus kinases (JAKs) or other tyrosine kinases (*Philips et al., 2022*). The clinical relevance of hyperactive STAT3 has been linked to subsets of hematological malignancies, with the identification of JAK1/3 or STAT3 mutations (*Jerez et al., 2012*; *Koskela et al., 2012*; *Crescenzo et al., 2015*; *Milner et al., 2015*). The most common STAT3

mutations, Y640F and D661Y, render STAT3 constitutively active (*Jerez et al., 2012*; *Koskela et al., 2012*). In sharp contrast, STAT3 mutations rarely occur in solid tumors. TCGA pan-cancer analysis reveals that most cancers do not express high levels of activated STAT3 (https://www.cancer.gov). Patient-derived xenografts and genetically engineered mouse models have yielded contrasting findings regarding the role of STAT3 in cancer development that range from tumor-promoting to tumor-suppressive, suggesting a high degree of tissue specificity (*Huynh et al., 2019*).

We aim to delineate the role of STAT3 in shaping the patterns of oncogenic KRAS dependency in KRAS mutant cancer cells. A previous study from our laboratory uncovered a novel link between STAT3 and cancer showing that activation of STAT3 in KRAS mutant cancers led to the stabilization of epithelial differentiation (*D'Amico et al., 2018*). This observation suggests that STAT3 plays a dynamic role in modulating the phenotypic diversity of KRAS-driven tumors, ostensibly coupled with the selection of the fittest variants. In this study, we leverage isogenic STAT3 intact and deficient cells to more fully delineate the effects of STAT3 on oncogenic KRAS dependency and the growth of cancer cells in culture or as tumors. To determine whether KRAS-dependent tumor cells are co-dependent on STAT3, we used two wellestablished models of KRAS mutant cancer: mouse embryonic fibroblasts and pancreatic ductal adenocarcinoma (PDAC) cells expressing endogenous KRAS$^{G12D}$. Both in vitro and in vivo assays demonstrate that neither persistent activation of STAT3 nor its loss confers distinct growth advantages on tumor cells. Instead, STAT3 guides morphological and functional characteristics of the transformed cells and tumors. Stabilization of the epithelial phenotype and attenuation of the TGF-β/SMAD4 pathway are two main driving forces behind STAT3 activity (*Oft et al., 1996*; *David and Massagué, 2018*; *Principe et al., 2021*; *Gough et al., 2021*). The data highlight antagonistic epistasis between SMAD4 and STAT3, where SMAD4 expressing tumors are poorly differentiated and exhibit mesenchymal features only in the absence of STAT3, while SMAD4-deficient tumors are well-differentiated and display epithelial morphology only in presence of STAT3. The results have implications for our understanding of the molecular basis of oncogenic KRAS dependency and therapy response.

## Results
### Effect of STAT3 activity on KRAS-mediated transformation

To assess the role of STAT3 in KRAS-driven tumorigenesis, we measured proliferation rates, contact inhibition, and tumor formation in mice. We have reported that p53-null mouse embryonic fibroblasts expressing endogenous mutant *Kras$^{G12D}$* (termed KP MEFs) exhibit typical features of oncogenic transformation using quantitative and sensitive assays (*Ischenko et al., 2013*). CRISPR/Cas9-mediated gene editing was used to generate isogenic STAT3 knockouts in the KP MEFs, and gain-of-function (GOF) and loss-of-function (LOF) mutant STAT3 alleles were stably integrated into cells via lentiviral vectors (*Figure 1A and B*). We used naturally occurring GOF mutants, Y640F, K658Y and D661Y, and a synthetic mutant STAT3C (A662C/N664C), which all render a persistently phosphorylated and thus hyperactive STAT3 pY705 (*Crescenzo et al., 2015*; *Bromberg et al., 1999*). LOF mutations targeted the STAT3 DNA binding (EE434-435AA and VVV461-463AAA) and transactivation domains (Y705F and S727A; *Horvath et al., 1995*; *Figure 1—figure supplement 1*). All mutants exhibited relatively uniform expression levels that were ~fivefold higher than endogenous STAT3 (*Figure 1B*, *Figure 1—figure supplement 1*). As reported, STAT3 Y640F, K658Y, and STAT3C displayed increased levels of Y705 phosphorylation. As neither STAT3 WT nor STAT3 GOFs exhibited robust phosphorylation on S727, we used a validated STAT3 mutant S727E that mimics the phosphorylation of S727 (*Qin et al., 2008*; *Figure 1C*, *Figure 1—figure supplement 1*).

Isogenic KP MEF cell lines harboring wild-type or mutant STAT3 were evaluated for growth, tumorigenesis, and pathway activation. Loss of STAT3 expression did not affect cell growth under standard culture conditions (*Figure 1—figure supplement 1*). Likewise, loss of STAT3 did not affect KRAS-induced transformation. This is indicated by the ability of STAT3 KO cells to grow in multilayers and form transformed foci to the same extent as controls (*Figure 1C*). Similar results were obtained using LOF mutations in STAT3 DNA binding (EE434-435AA and VVV461-463AAA) and transactivation domains (Y705F and S727A). In contrast, STAT3 GOF mutations, Y640F, D661Y, and K658Y, impaired KRAS-induced focus formation (p<0.005 by two-tailed T test, *Figure 1C and D*). Because the transduced STAT3 constructs co-expressed a GFP reporter, the formation of transformed foci was

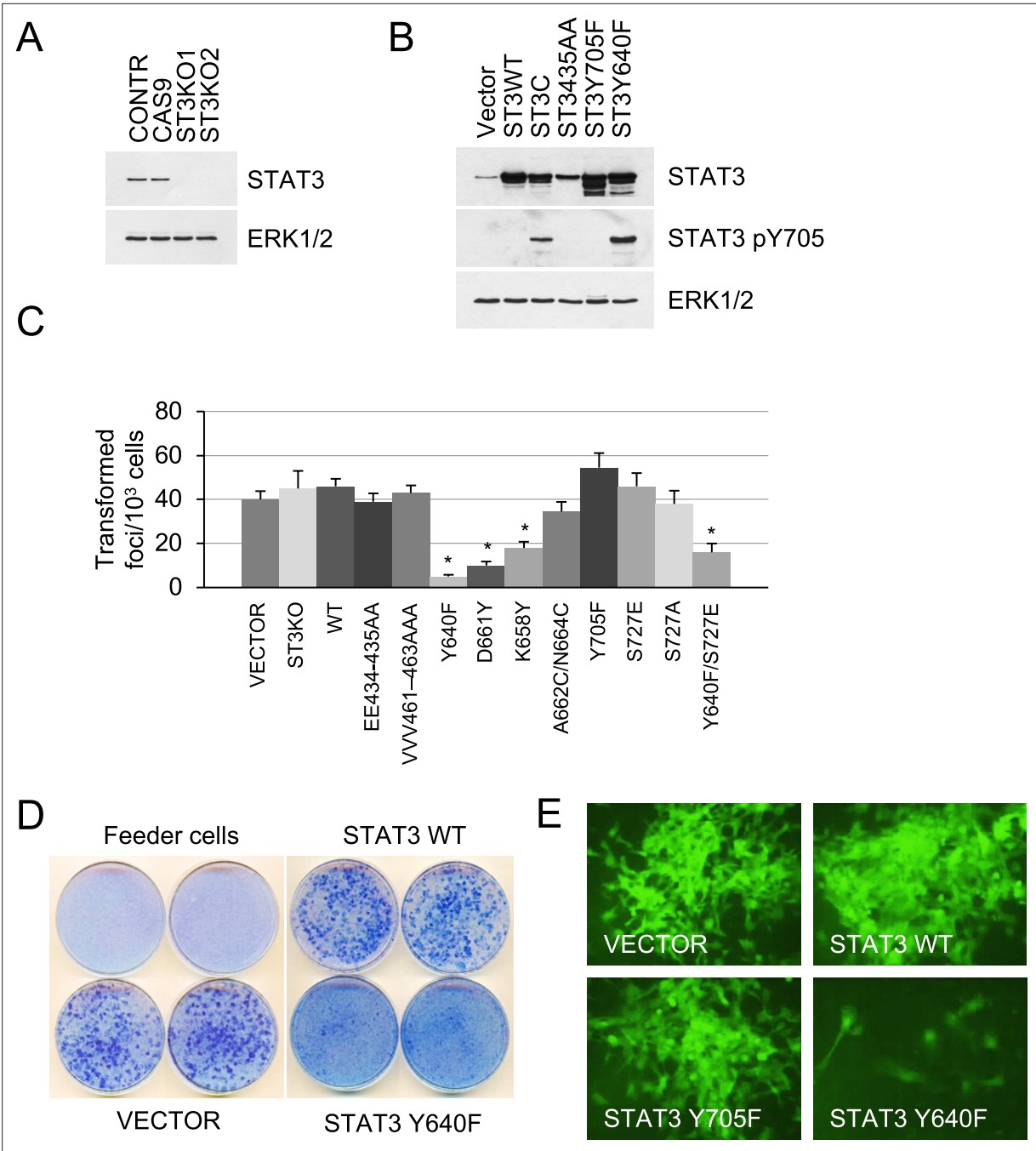

**Figure 1.** Effect of STAT3 activity on KRAS-mediated transformation. (**A**) Western blot analysis of STAT3 expression in KP MEFs (Contr) transduced with CRISPR/CAS9 lentivirus expressing CAS9 or gRNAs targeting STAT3 (ST3KO). Two independent knockout clones are shown. ERK1/2 is a loading control. (**B**) Western blot analysis of total and Y705-phosphorylated STAT3 in KP MEFs transduced with lentivirus expressing control (Vector), wild-type (ST3WT) or mutant STAT3 alleles as noted. ERK1/2 is a loading control. (**C**) Transduced control (Vector) and KP MEFs were evaluated for foci formation. Cells expressing the indicated STAT3 genotypes were co-cultured with $10^3$ p53KO feeder MEFs and macroscopic colonies were counted after 2 weeks (n=3 with six replicate plates for each cell type, *p<0.05). Values correspond to mean ± s.d. (**D**) Representative images of tissue culture plates stained with Giemsa to detect foci formed by KP MEFs expressing vector, STAT3 WT or hyperactive STAT3 Y640F. (**E**) Representative images of transformed foci visualized by fluorescence microscopy. Foci formed by KP MEFs co-expressing GFP with vector alone, STAT3 WT, STAT3 Y705F, or STAT3 Y640F are shown.

The online version of this article includes the following source data and figure supplement(s) for figure 1:

**Source data 1.** Original western blot images.

**Figure supplement 1.** STAT3 mutant alleles and their effect on growth in culture.

*Figure 1 continued on next page*

Figure 1 continued

**Figure supplement 1—source data 1.** Original western blot images.

**Figure supplement 2.** Analysis of STAT3 loss-of-function and gain-of-function.

visualized by fluorescence microscopy (**Figure 1E**). Through these real-time studies, we discovered that only a small percentage of cells with hyperactive STAT3 Y640F had some ability to form transformed colonies, while the majority of cells remained contact inhibited. We noted that STAT3 S727E had no significant effect on cell transformation, while the Y640F/S727E double mutant displayed only a marginal increase in the number of transformed foci relative to STAT3 Y640F itself (**Figure 1C**). Thus, the hyperactive Y640F mutation exerts a dominant influence over S727E in KRAS-transformed MEFs. As expected, lentiviral expression of wild-type or GOF STAT3 alleles, Y640F and D661Y, failed to transform primary p53KO MEFs or immortalized NIH 3T3 cells (**Figure 1C and D**; **Figure 1—figure supplement 2**; data not shown), indicating that STAT3 does not display intrinsic oncogenicity on its own.

## STAT3 GOF mutations reduce tumor development in mice

To investigate tumorigenic effects of STAT3 in vivo, subcutaneous implants of KP MEFs into nude mice were used. Tumors developed by STAT3 KO cells showed growth characteristics similar to those of STAT3 intact vector controls (**Figure 2A**). In contrast, cells expressing the hyperactive STAT3 Y640F and, to a lesser extent, K658Y mutations were delayed in their ability to form tumors in mice. We used fluorescence-activated cell sorting for GFP to fractionate STAT3 Y640F MEFs into pools with low and high STAT3 pY705 expression (**Figure 2B**). Following implantation in mice, cell populations with high STAT3 pY705 expression developed tumors more slowly compared to low expressing cells (p=0.005, **Figure 2C**). Therefore, there is a dose-dependent ability of hyperactive STAT3 Y640F to limit tumorigenicity of KRAS-transformed MEFs. Limiting dilution assays in nude mice revealed that the frequency of tumor-initiating cells was reduced by approximately ninefold in high STAT3 Y640F expressing cells compared to control cells (**Figure 2D**).

To determine whether specific Y705 phosphorylation and DNA binding are required for STAT3 Y640F to exert its suppressive activity, we generated three double mutants: a phosphorylation defective STAT3 Y640F/Y705F, and two DNA-binding domain (DBD) mutants; STAT3 Y640F/R382W and Y640F/V463Δ (**Figure 1—figure supplement 2**). Both R382W and V463Δ are recurrent STAT3 mutations observed in humans (**Jiao et al., 2008**). A STAT3-responsive luciferase reporter assay was used to confirm that the DBD mutants are impaired in their ability to induce STAT3-mediated gene transcription. Notably, the DBD mutants of STAT3 Y640F lost the ability to attenuate KRAS-mediated MEF transformation despite their continuous pY705 phosphorylation (**Figure 2E**; **Figure 1—figure supplement 2**). The phosphorylation-defective STAT3 Y640F/Y705F double mutant was likewise impaired. We conclude that STAT3 Y640F-mediated inhibition of tumor development and KRAS-induced MEF transformation is dependent on STAT3 phosphorylation at Y705, DNA binding, and gene-specific transactivation.

To elucidate the means by which hyperactive STAT3 suppresses KP MEF transformation, we evaluated pathway activity by western blot analysis and whole exome RNA sequencing (RNA-seq). Western blot analyses of control and STAT3 Y640F-expressing MEFs showed unperturbed RAS/MAPK and PI3K/AKT signaling (as assessed by phosphorylated ERK1/2 and AKT1), suggesting that STAT3 does not directly alter these downstream KRAS effectors (**Figure 1—figure supplement 1**). RNA-seq analysis showed that expression of STAT3 Y640F in MEFs resulted in the differential expression of approximately 290 genes (p<0.05) compared to control cells. Gene ontology (GO) classification of biological processes showed an enrichment of pathways consistent with the role of STAT3 as a mediator of immunity and the inflammatory response (**Figure 1—figure supplement 2**). Biological processes attenuated by hyperactive STAT3 included differentiation and tissue development, and pathways mediated by TGF-β signaling (**Oft et al., 1996**; **David and Massagué, 2018**; **Zhang et al., 2017**). We therefore tested whether inactivation of the TGF-β pathway could inhibit KRAS-induced MEF transformation. Indeed, ablation of the TGF-β signaling components TGFBR2 or SMAD4 in KP MEFs using CRISPR/Cas9-mediated gene editing nearly eliminated foci formation and effects of STAT3 (**Figure 1—figure supplement 2**). The results support the premise that hyperactive STAT3 interferes with KRAS-induced

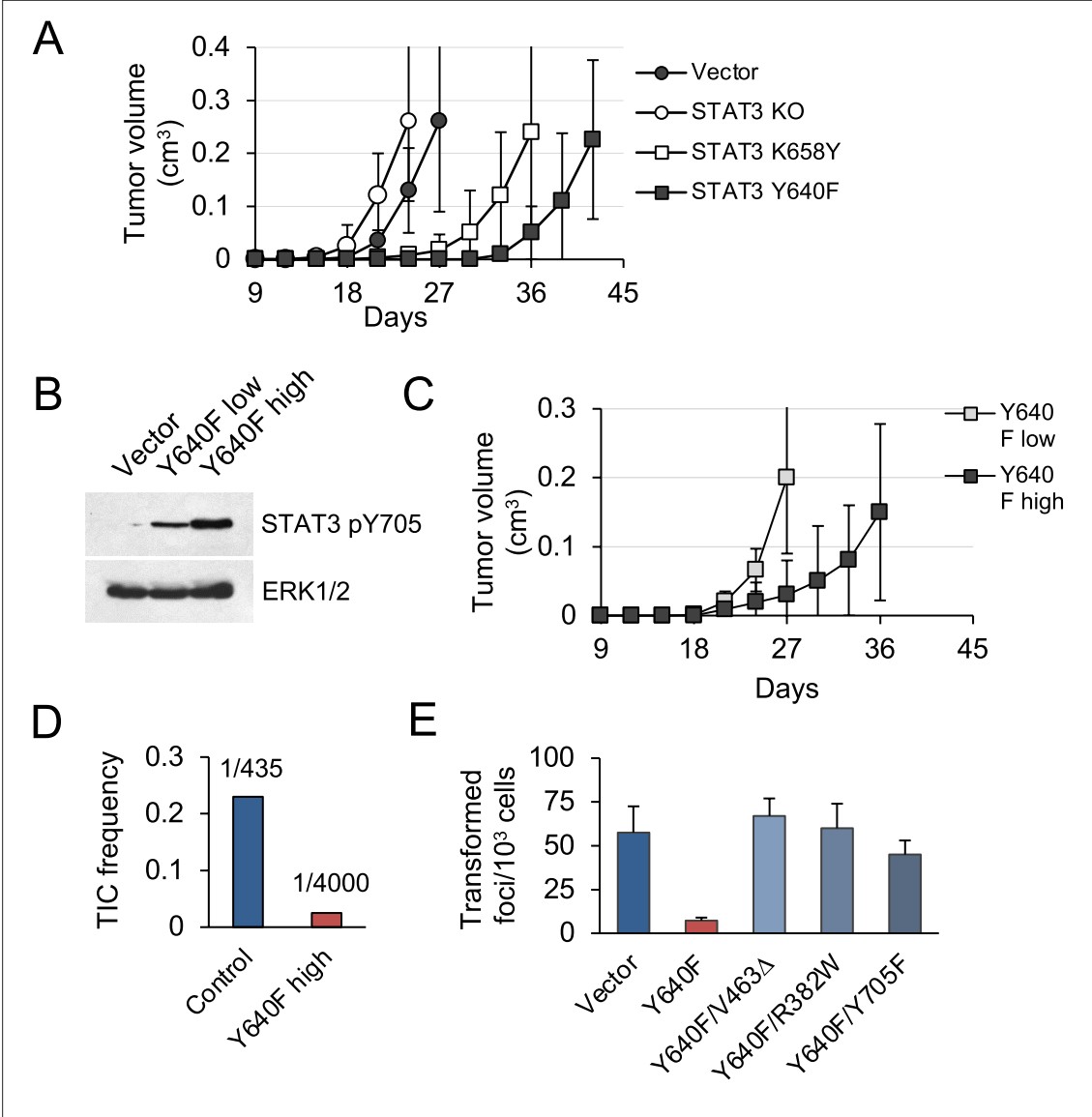

**Figure 2.** STAT3 GOF mutations reduce tumor development in mice. (**A**) Subcutaneous tumor formation in nude mice by KP MEFs expressing vector alone, STAT3 KO, STAT3 K658Y or STAT3 Y640F ($10^4$ cells per injection site, n=4 for each cell type). Error bars represent s.d. (**B**) Western blot analysis of Y705-phosphorylated STAT3 in KP MEFs expressing vector alone, and low or high levels of STAT3 Y640F protein. Cells were fractionated by FACS for GFP. ERK1/2 is a loading control. (**C**) Subcutaneous tumor formation in nude mice by cells from (**B**) ($10^4$ cells per implant site, n=4 for each cell type). Error bars represent s.d. (**D**) Quantification of tumor-initiating cell (TIC) frequency in control (Vector) and STAT3 Y640F-expressing KP MEFs by extreme limiting dilution assays (ELDA) in nude mice. (**E**) Focus formation by KP MEFs expressing vector alone, STAT3 Y640F or double mutants of STAT3 Y640F with DNA binding domain (DBD) mutations, V463Δ and R382W, or phospho-inactivating Y705F. Results represent replicates from two independent experiments (n=4 for each cell type). Values correspond to mean ± s.d.

The online version of this article includes the following source data and figure supplement(s) for figure 2:

**Source data 1.** Original western blot images.

**Figure supplement 1.** Gene regulation in response to loss of STAT3, SMAD4, and mutant KRAS in PDAC.

**Figure supplement 1—source data 1.** Original western blot images.

transformation through suppression of the TGF-β pathway. This prompted us to investigate the functional interaction of STAT3 and TGF-β/SMAD4 in epithelial carcinogenesis.

## STAT3 is a genetic modifier of EMT

A notable feature of KRAS mutant cancers, including those of the pancreas, colon, and lung, is that they tend to fall into two classes based on their canonical KRAS and TGF-β signaling; those that have a strong dependence on KRAS signaling (KRAS-dependent) or those that have less dependence on canonical KRAS signaling (KRAS-independent; *Singh et al., 2009*; *Yuan et al., 2018*). KRAS-dependent tumors have been associated with an epithelial gene signature and morphology, whereas KRAS-independent tumors show enriched expression of mesenchymal genes. Since STAT3 and TGF-β have been shown to compete, cooperate, or antagonize each other in many other contexts (*Wang et al., 2016*; *Babaei et al., 2018*; *Biffi et al., 2019*; *Jenkins et al., 2005*), we investigated STAT3 as it relates to KRAS dependency. To that end, we used murine pancreatic ductal adenocarcinoma (PDAC)-derived cell lines bearing endogenous $Kras^{G12D}$ and $Tp53^{R172H}$ mutations (termed KPC) (*Hingorani et al., 2005*). CRISPR/Cas9 gene editing was used to ablate STAT3, SMAD4, or TGFBR2 expression in PDAC cells (*Figure 2—figure supplement 1*) and the behavior of these cells was tested alongside previously generated KRAS knockout cells (*Ischenko et al., 2021*). The cells were implanted orthotopically into the pancreata of nude mice, and animals were observed for latency of tumor formation and changes in tumor morphology.

Pancreatic tumors were detected within 3 weeks following implantation of $10^4$ parental control, STAT3 knockout (KO), SMAD4 KO, or TGFBR2 KO PDAC cells, and there was no statistical difference in tumor latency between the groups. However, there was considerable difference in tumor morphology. Tumors in the parental (intact) group were characterized by classical adenocarcinoma-like morphology with glandular structures (*Figure 3A*). In comparison, KRAS KO tumors displayed a highly sarcomatoid morphology indicative of full EMT. Loss of STAT3 also induced a morphological change compatible with EMT, whereas overexpression of hyperactive STAT3 Y640F resulted in the sporadic co-occurrence of squamous and glandular differentiation. Loss of SMAD4 or TGFBR2 was associated with increased epithelial differentiation relative to controls. This is consistent with previous findings showing that the TGF-β pathway is a key regulator of tumor cell differentiation and malignant behavior, but not growth rate (*Izeradjene et al., 2007*; *Dai et al., 2021*).

We determined whether differences in tumor morphology were coordinate with changes in gene expression. Comparative RNA-seq analyses of parental KPC control and STAT3 KO cell lines revealed distinct transcriptional profiles that included more than 700 differentially expressed genes (*Figure 3B*). The enriched genes in KPC parental cells included those corresponding to the major structural proteins in epithelial cells, such as cadherins, claudins, and tight junctions (*Figure 3C*). In contrast, STAT3 KO cells were enriched in signatures of EMT and embryonic organ morphogenesis (*Figure 3D*). The overall pattern of gene expression suggests that loss of STAT3 is associated with activation of partial rather than complete EMT, since epithelial markers (e.g. CDH1 and EpCAM) continue to be expressed, but mesenchymal markers (e.g. FN1 and various collagens) have been acquired (*Brabletz et al., 2018*; *Lambert and Weinberg, 2021*; *Figure 2—figure supplement 1*). As TGF-β classically promotes EMT, we also focused on TGF-β family genes (*Korkut et al., 2018*). Among these genes, STAT3 KO cells had a significant increase in TGFB1, TGFB3 and INHBA expression (*Figure 2—figure supplement 1*). The expression of EMT-activating transcription factors SNAI, TWIST and ZEB was not strongly affected, indicating that induction of EMT involves additional STAT3 dependent regulators. We did identify transcription factors, such as JUNB and SOX4, that associate with EMT (*David et al., 2016*). We computed EMT scores using gene expression values of epithelial (EPI) and mesenchymal (MES) genes. STAT3 and KRAS KO KPC cells displayed similar levels of EMT at the level of gene expression (*Figure 2—figure supplement 1*; *Serresi et al., 2021*). In contrast, SMAD4 KO cells displayed reduced expression of EMT-related genes, while genes involved in epithelial differentiation and RAS dependency were among the most upregulated.

Notably, comparative analysis between STAT3 and KRAS knockout KPC cells revealed approximately 250 STAT3 target genes (>30%) that were similarly up- or down-regulated (Pearson's $r$=0.88, p<0.00001), suggesting that STAT3 loss partially phenocopies the effects of KRAS inactivation (*Figure 3E*). GO classifications of the overlapping genes in KRAS and STAT3 knockouts included developmental processes and mesenchymal tissue remodeling (*Figure 3D*; *Figure 2—figure supplement*

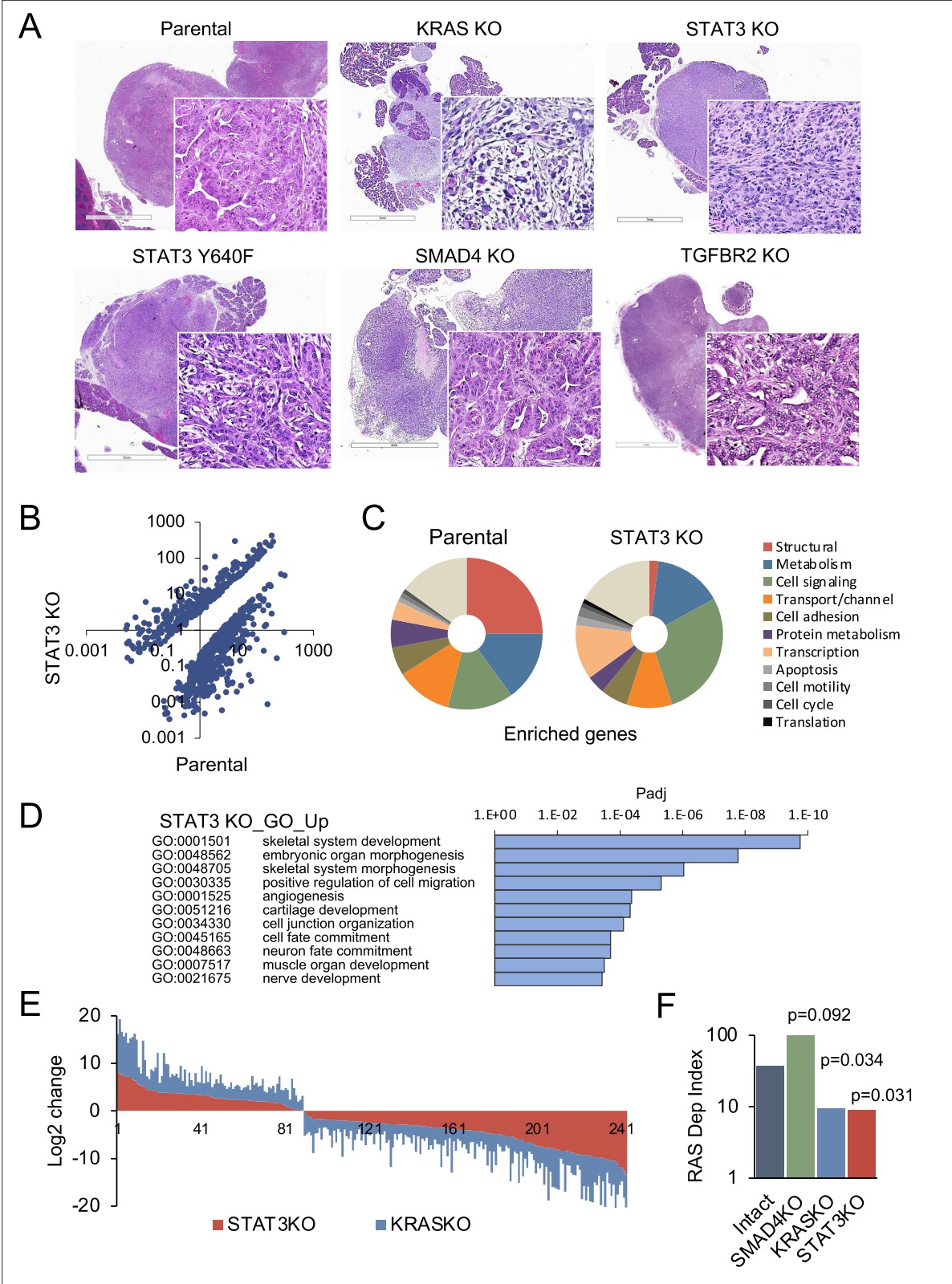

**Figure 3.** STAT3 is a genetic modifier of EMT. (**A**) Representative H&E staining of pancreatic tumor sections derived from KPC cells of the indicated KRAS (n=15'), STAT3 (n=11), TGFBR2 (n=3), and SMAD4 (n=12) genotypes, where n denotes the number of mice injected. At least two independent clones were used. 2mm scale bar. (**B**) Scatter plot representing differentially expressed genes in parental (STAT3 intact) and STAT3 KO KPC cells. (**C**) Pie charts representing gene expression profiles of the top 100 upregulated genes in parental control (STAT3 intact) and STAT3 KO KPC cells with

*Figure 3 continued on next page*

*Figure 3 continued*

percentages linked to specific cellular functions. (**D**) GO enrichment analysis of the top 100 upregulated genes in STAT3 KO cells compared to STAT3 intact (control) cells. (**E**) Analysis of overlapping differentially expressed genes (n=243) in STAT3 KO and KRAS KO KPC cell lines relative to parental control cells. (**F**) Mouse RAS dependency index (RDI) of KPC cells of the indicated KRAS, STAT3, and SMAD4 genotypes.

*1*). As a proof of concept for the functional connection of STAT3 to KRAS dependency, we used gene expression data to compute RAS dependency scores for parental and knockout cells. A mouse KRAS dependency signature (21 genes) was derived from single cell RNA-seq data of KRAS intact vs. KRAS knockout PDAC tumors and used for the analyses (*Figure 2—figure supplement 1*; *Ischenko et al., 2021*). Results showed RAS dependency scores had a significant positive correlation with STAT3 and KRAS expression, and a negative correlation with SMAD4 expression (*Figure 3F*). Whole tumor RNA-seq analysis of STAT3 intact and knockout pancreatic tumors supported these findings (*Figure 2—figure supplement 1*). Together these data indicate that STAT3 is a genetic modifier that can regulate KRAS dependency and tumor development through counterposing EMT.

## STAT3 and SMAD4 play opposing roles in pancreatic tumorigenesis

The apparent antagonism between STAT3 and induction of EMT was of particular interest, as TGF-β-induced EMT appears to confer adaptive resistance to KRAS inhibition (*Singh et al., 2009*; *Hou et al., 2020*). To test the relative importance of the STAT3 and TGF-β pathways to tumor morphology and functionality, we generated STAT3/SMAD4 and STAT3/TGFBR2 double knockout (DKO) cell lines. DKO cell lines formed pancreatic tumors in mice, but their histological features were distinct from SMAD4 or TGFBR2 single knockouts as they produced mixed epithelial/mesenchymal morphologies with cells expressing both E-cadherin and vimentin (*Figure 4A and B*). SMAD4 or TGFBR2 intact tumors displayed features of EMT in the absence of STAT3, whereas SMAD4 or TGFBR2 KO tumors displayed a well-differentiated epithelial phenotype only in the presence of STAT3 (*Figures 3A and 4A*). The data reinforce the notion that functional antagonism of STAT3 and TGF-β/SMAD4 controls PDAC development and KRAS dependency (*Figure 4C*).

We analyzed human PDAC databases to determine if our STAT3 and SMAD4 gene expression signatures could be projected onto human tumors. SMAD4 and STAT3 signature scores were computed from the top up- or down-regulated genes in KPC cell lines (*Figure 4—source data 1*). When aligned with human PDAC samples from the TCGA cohort (stage I/II tumors), STAT3 and SMAD4-regulated gene signatures demonstrated significant statistical correspondence, supporting the selective antagonism of STAT3 and SMAD4 (Pearson's $r>0.5$, $p<0.00001$; *Figure 4D*). As SMAD4 is frequently deleted in PDAC, samples expressing only wild-type SMAD4 were manually curated from the TCGA cohort (n=112). Tumors were classified as either epithelial (EPI) or mesenchymal (MES) using previously characterized gene sets (*Figure 4—source data 2*). Results demonstrate that STAT3-regulated gene expression is closely associated with epithelial differentiation, while SMAD4-regulated gene expression is strongly associated with EMT (Pearson's $r>0.7$) (*Figure 4E*). The association with KRAS dependency status was also revealing. PDAC tumor samples from the TCGA cohort were grouped as KRAS-dependent/KRAS type or KRAS-independent/RSK type based on previously derived KRAS dependency signatures (*Figure 4—source data 3*; *Singh et al., 2009*; *Yuan et al., 2018*). KRAS-dependent tumors showed enriched expression of epithelial genes (EPI) and reduced expression of mesenchymal genes (MES), whereas the KRAS-independent samples displayed the inverse, as previously reported (*Singh et al., 2009*; *Yuan et al., 2018*; *Tan et al., 2014*). Importantly, the KPC STAT3 knockout gene signature (i.e. upregulated genes) co-segregated with human KRAS-independent/mesenchymal PDAC tumors, while the SMAD4 knockout gene signature (i.e. upregulated genes) co-segregated with human KRAS-dependent/epithelial tumors (*Figure 4—figure supplement 1*). A similar trend was observed for the PanCuRx Translational Research Initiative (COMPASS, stage IV PDAC) cohort mainly composed of liver metastases, as the STAT3-reliant gene signature was also enriched in epithelial and KRAS dependent samples ($r>0.7$, $p<0.00001$; *Figure 4E*). Liver is the main site of PDAC metastases, and pancreatic cancer metastases commonly display a stabilized epithelial phenotype (*Carstens et al., 2021*; *Reichert et al., 2018*). Overall, results demonstrate that STAT3-regulated gene expression is closely associated with epithelial differentiation and RAS dependency, while SMAD4-regulated gene expression is strongly associated with EMT and RAS independence.

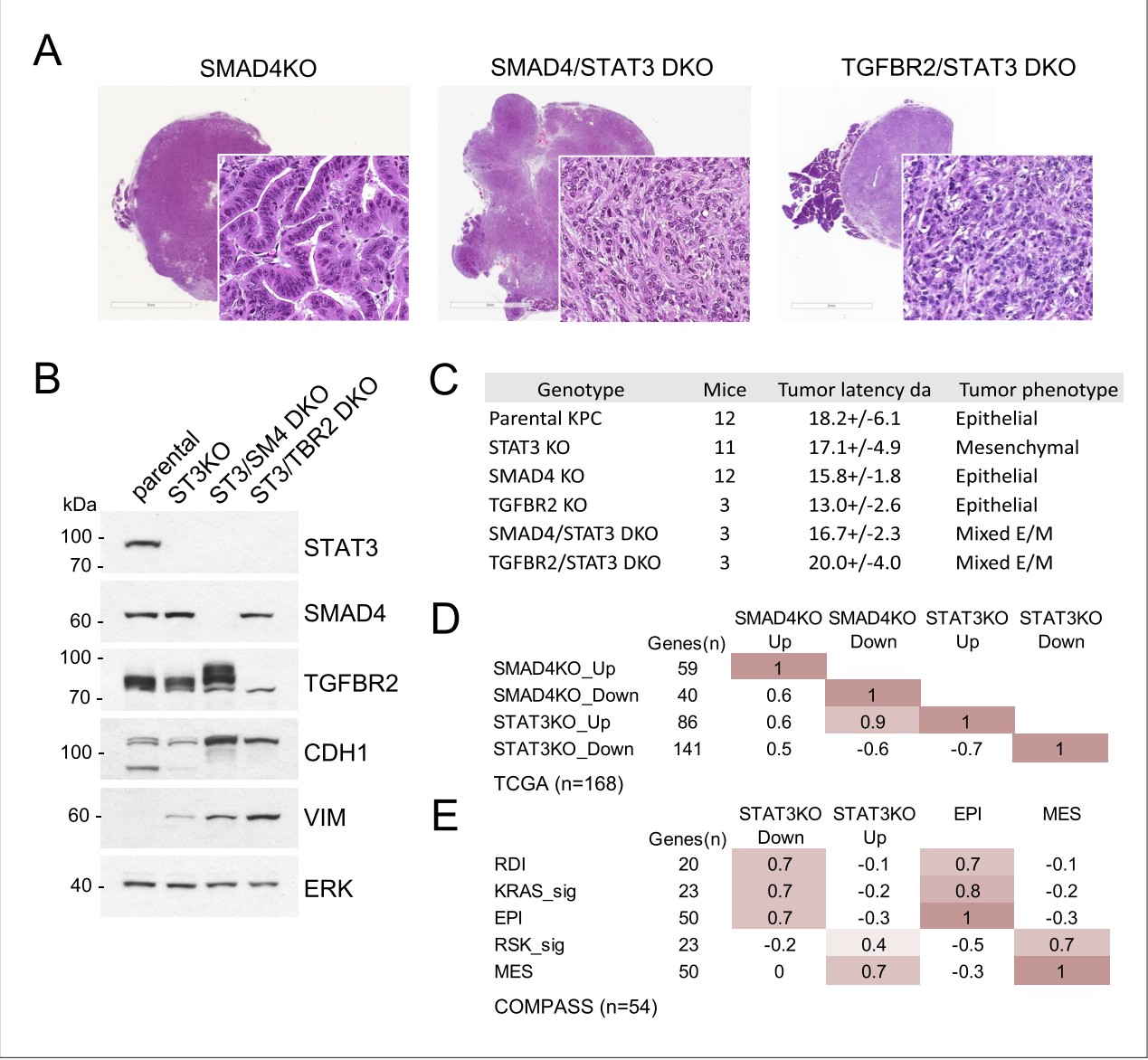

**Figure 4.** STAT3 and SMAD4 play opposing roles in pancreatic tumorigenesis. (**A**) Representative H&E staining of pancreatic tumor sections derived from KPC cells of the indicated genotypes, SMAD4 KO (n=12), SMAD4/STAT3 DKO (n=3) and TGFBR2/STAT3 DKO (n=3), where n denotes the number of mice injected. At least two independent clones were used. 2mm scale bar. (**B**) Western blot analysis of control (Intact), STAT3 KO (ST3KO), STAT3/SMAD4 double KO (ST3/SM4DKO), or STAT3/TGFBR2 double KO (ST3/TRB2DKO) KPC cells. Expression of STAT3, SMAD4, TGFBR2, CDH1 (E-cadherin), and VIM (Vimentin) is shown. ERK1/2 is a loading control. (**C**) Summary of pancreatic tumor development in nude mice by KPC cells of the indicated STAT3, SMAD4, and TGFBR2 genotypes presented. (**D**) Pearson correlation heatmap comparing gene expression signatures of SMAD4 KO (SMAD4KO_up, down) and STAT3 KO (STAT3KO_up, down) KPC cells with human PDACs from TCGA (n=168). Signature scores were calculated using the top up- and down-regulated genes in STAT3 and SMAD4 KO KPC cells. The number of genes for each comparison is shown. (**E**) Pearson correlation heatmap comparing gene expression signature scores of genes regulated in STAT3 KO KPC cells with pancreatic tumors from the COMPASS database (n=92) classified by RAS dependency index (RDI), KRAS-dependent signature (KRAS-sig), KRAS-independent signature (RSK-sig), epithelial (EPI), or mesenchymal (MES) gene signature score.

The online version of this article includes the following source data and figure supplement(s) for figure 4:

**Source data 1.** Top up- or down-regulated genes in murine KPC STAT3 KO or SMAD4 KO cells derived from RNA-Seq.

**Source data 2.** Signature genes used to designate human PDAC tumors from TCGA and COMPASS databases as either epithelial (EPI) or mesenchymal (MES).

**Source data 3.** Genes used to define a RAS Dependency Index (RDI) derived from A.

**Figure supplement 1.** Human pancreatic cancer expression of STAT3 and SMAD4.

These findings underscore our basic premise that there exists an epistatic antagonism between STAT3 and SMAD4, highlighting a new role for STAT3 as a genetic modifier in KRAS mutant cancer.

## Discussion

The findings presented in this study have significant implications in two main aspects. We provide evidence that the STAT3 transcription factor acts as a genetic modifier of EMT and KRAS dependency in a mouse model of pancreatic carcinogenesis. Our results demonstrate that neither persistent activation nor genetic ablation of STAT3 confers a selective growth advantage on tumor cells. Instead, STAT3 plays a crucial role in guiding the morphological and functional characteristics of tumors by inhibiting EMT and maintaining epithelial identity. Furthermore, our study uncovers an intriguing relationship between the SMAD4 and STAT3 transcription factors in PDAC. While the involvement of STAT3 in cancer has been predominantly associated with chronic inflammation and fibrosis (*Laklai et al., 2016*; *Ji et al., 2019*), our data underscore the significance of epistasis as a key factor that underlies functional antagonism between STAT3 and TGF-β/SMAD4 signaling. Our findings shed light on the regulatory mechanisms involved in cancer progression.

Oncogenic KRAS mutations are observed in approximately 90% of pancreatic cancers and less frequently in other cancer types. However, the role of KRAS in PDAC maintenance, once held to be nearly absolute, has shown limitations. A notable feature of KRAS mutant cancers, including those of the pancreas, colon, and lung, is that they can be either KRAS-dependent or KRAS-independent, based on the degree of their addiction to canonical KRAS signaling. The concept of KRAS dependency, originally introduced as a measure of oncogenic addiction following KRAS inactivation, has proven to be multifaceted. It integrates KRAS signaling outputs, with effector topologies, cooperating mutations, and environmental cues (*Singh et al., 2009*; *Yuan et al., 2018*; *Ischenko et al., 2021*; *Brubaker et al., 2019*). Cellular morphology (epithelial versus mesenchymal) appears to be one of the most noticeable manifestations of different degrees of KRAS dependency. Although SMAD4 classically promotes EMT and KRAS independence (*Singh et al., 2009*), approximately 50% of moderately to well-differentiated tumors in the TCGA cohort are, in fact, SMAD4 wild-type. This indicates that SMAD4 mutation alone is not sufficient to predispose cancer cells to a particular RAS phenotype. Our study emphasizes the mutual antagonism of SMAD4 and STAT3, and a fundamental role of STAT3 in the maintenance of epithelial cell identity. While SMAD4 wild-type tumors displayed features of EMT in the absence of STAT3, SMAD4 KO tumors displayed an epithelial phenotype only in the presence of STAT3. The results demonstrate that STAT3 and SMAD4 inversely contribute to oncogenic dependency. STAT3 sustains the KRAS-dependent phenotype and tumor aggressiveness, with the potential for improved efficacy of anti-RAS drugs, whereas SMAD4 promotes KRAS independence at the expense of enhanced therapy resistance. Therefore, the epistatic relationship between SMAD4 and STAT3 has implications for tumor aggressiveness, metastatic propensity, and therapeutic resistance.

Genes involved in cancer (~200 drivers validated to date) affect critical cellular processes, rendering them tumorigenic or tumor suppressive. The Cancer Dependency Map project sets out to model the genetic landscape of cancer in accordance with the oncogene addiction paradigm. By employing high throughput RNAi or CRISPR knockout screens across a multitude of cancer-derived cell lines, the goal is to broadly identify putative cellular dependencies for cancer therapy. While an important undertaking, not all cell lines align well with tumor samples in terms of mutations and gene expression profiles (*Warren et al., 2021*). STAT3 exemplifies this problem, as efforts to understand its role of STAT3 in cancer have resulted in conflicting reports that show either a positive or negative role in tumor development (*Yu et al., 2014*; *Huynh et al., 2019*). Large-scale analysis of patient-derived PDACs (n=84) and pancreatic cell lines from the Cancer Cell Line Encyclopedia (n=39) reveal low to medium levels of STAT3 Y705 phosphorylation (*Figure 4—figure supplement 1*). Endogenous phosphorylation/activation of STAT3 appears able to function within a narrow operating range in multiple solid tumor types.

Cancers are complex biological systems exhibiting inexplicable levels of intractability and unpredictability. For instance, even when challenged with the same lethal anticancer drugs used in vitro, cancers show remarkable resistance in vivo. Further complicating the analysis, there exist KRAS mutant cell lines whose survival and growth are no longer dependent on continued KRAS activity (*Ischenko et al., 2021*; *Muzumdar et al., 2017*; *Lentsch et al., 2019*). This raises doubts regarding their eventual responsiveness to targeted anti-KRAS therapies. Here, we used genetic analyses to

identify STAT3 as a relevant dependency in KRAS-driven cancer. While in vitro evidence indicates that STAT3 lacks classical driver properties, it nevertheless plays an essential role in cancer maintenance and epithelial-mesenchymal plasticity. This may explain the rarity of STAT3 GOF mutations in human cancers with mutant KRAS.

## Materials and methods

### Cells and reagents

Clonally-derived $Kras^{G12D}$ $Tp53KO$ mouse embryonic fibroblasts (KP MEFs) and pancreatic $Kras^{G12D}$ $Tp53^{R172H}$ (KPC) cells were previously described (*Ischenko et al., 2013*; *Hingorani et al., 2005*). Cells were authenticated by partial sequencing and were tested to be free of mycoplasma. Human Hep3B and HEK293T cells were obtained from ATCC. All cells were grown in DMEM media supplemented with 5% FBS (Atlanta Biologicals) and 1 x antibiotic/antimycotic (Gibco). For standard proliferation assays, cells were seeded into six-well plates at a concentration of $4x10^5$ cells per well and counted cumulatively with a Coulter counter (Beckman) every 3 days for 2 weeks. Focus formation assays were performed as described (*Ischenko et al., 2013*; *Ischenko et al., 2014*). Briefly, $10^3$ KP MEFs were co-cultured with $10^5$ p53KO feeder MEFs in 6 cm dishes. After 2 weeks, transformation efficiency was evaluated by manually counting macroscopic colonies. Transformed foci were stained with Giemsa (Sigma) for visualization. Stable KP MEF and KPC knockout cell lines were generated via selection in 2 μg/ml puromycin followed by single colony isolation. In vitro luciferase activity was measured using a Lumat model LB luminometer (Promega) and the Luciferase Reporter Gene Assay according to manufacturer's instructions (Roche). Hep3B cells were stimulated with human IL-6 (Cell Signaling Technology) at a concentration of 20 ng/ml for 24 hr.

### Lentivirus and plasmids

Lentiviral (adapted from the pWPXL/pEF1a backbone) and pEGFP-N1 (Addgene) expression vectors encoding mutant STAT3 alleles were derived using site-directed mutagenesis. The final plasmids were sequence confirmed. The p3XGAS-Hsp70-Luc reporter plasmid was described (*Foreman et al., 2017*). For CRISPR/Cas9-mediated knockouts, we used single guide RNAs (sgRNAs) for *Stat3* (5'-gcag ctggacacacgctacc-3' or 5'-gtacagcgacagcttccca-3'), *Smad4* (5'-ggtggcgttagactctgccg-3'), and *Tgfbr2* (5'-ccttgtagacctcggcgaag-3') cloned into LentiCRISPRv2 puro (Addgene) (*Sanjana et al., 2014*). Lenti-viruses were produced by transient transfection of HEK293T and collected according to standard protocols.

### Tumorigenicity in mice

All animal studies were approved by the Institutional Animal Care and Use Committee at Stony Brook University. Male NU/J (nude) mice (5 weeks old) (The Jackson Laboratory) were inoculated orthot-opically or subcutaneously with $10^4$ cells in 100 μl of Matrigel (Corning) diluted 1:7 with Opti-MEM (Corning). Orthotopic implantations into the pancreas were performed using standard procedures (*López-Novoa and Nieto, 2009*). Pancreatic tumor latency was determined through abdominal palpa-tion. We defined subcutaneous tumor latencies as the period between implantation of tumorigenic cells into mice and the appearance of tumors 1 mm in diameter. The end point was a tumor diameter of 0.5 cm. Statistical analyses were performed using two-tailed Student's t-test at the 95% confi-dence interval. $p \leq 0.05$ was considered statistically significant. Tumor-initiating cell (TIC) frequency was determined by extreme limiting dilution assays and online ELDA software (https://bioinf.wehi.edu.au/software/elda/). The number of cells in each subcutaneous injection ranged from $10^2$ to $10^4$. Mouse tumor tissue was harvested, immersion fixed in five volumes of 4% paraformaldehyde for 48 hr, and processed via the Stony Brook University Histology Core. Paraffin-embedded formalin-fixed 5 μm sections were stained with hematoxylin and eosin for histology.

### Expression analysis

Western blotting was performed using antibodies against AKT1 (4691), P-AKT1 pS473 (4060), CDH1 (3195), P-ERK1/2 (4370), SMAD4 (46535), P-STAT3 pY705 (9131), VIM (5741) (all from Cell Signaling), ERK1/2 (05–157, Millipore), STAT3 (610190, BD) and TGFBR2 (sc-400, Santa Cruz). Whole cell extracts were prepared by lysing cells in buffer containing 10 mM Tris HCl, pH7.4, 150 mM NaCl, 1 mM EDTA,

10% glycerol, 1% Triton X100, 40 mM NaVO4, 0.1% SDS, and 1 x protease inhibitors (Roche). Western blots were imaged using Image Studio software (LI-COR). Total cellular RNA was isolated using Pure-Link RNA (Thermo Fisher) according to manufacturer's specifications and phenol-extracted. Pancreatic tissues were incubated for 24 hr at 4 °C in ≥5 volumes of RNAlater solution (Thermo Fisher) to preserve RNA integrity. RNA sequencing and bioinformatics were performed by Novogene Corporation (https://en.novogene.com). STAT3 and SMAD4 knockout signature scores (STAT3KO_UP, STAT3KO_DN, SMAD4KO_UP, SMAD4KO_DN) were computed as the average of RNA expression values (fpkm) of the top up- or down-regulated genes in KPC cell lines. EMT and mouse Ras Dependency Index (RDI) scores were calculated using defined gene sets (*Ischenko et al., 2021*; *Serresi et al., 2021*; *Tan et al., 2014*). For publicly available human datasets, RDI, KRAS_sig, RSK_sig, epithelial (EPI) and mesenchymal (MES) gene expression scores were calculated as the sum of RNA expression values (z-scores) using previously characterized gene modules (*Singh et al., 2009*; *Yuan et al., 2018*).

## Statistics and reproducibility

Statistical analysis was performed using two-tailed Student's t-test, Fisher's exact test or Wilcoxon test, as appropriate for the dataset. ARRIVE guidelines were used in the study as appropriate. An FDR adjusted p-value (q-value) was calculated for multiple comparison correction. Individual mice and tumor cell lines were considered biological replicates. Statistical details for each experiment are denoted in the corresponding figures and figure legends. The micrographs (H&E) represent at least three independent experiments. All data are presented as mean ± SD. In box and whisker plots, the middle line is plotted at the median, the upper and lower hinges correspond to the first and third quartiles, and the ends of the whisker are set at 1.5 x IQR above the third quartile and 1.5 x IQR below the first quartile (IQR, interquartile range or difference between the 25th and 75th percentiles).

## Acknowledgements

This work was supported by NIH grant RO1CA236389 and the Carol M Baldwin Breast Cancer Research Award to NCR, and the Catacosinos Cancer Research Award to OP. We wish to thank Fang Yuan Hao for his assistance, Jean Rooney in the Stony Brook University Division of Laboratory Animal Research for her technical assistance in mouse surgeries, and orthotopic implants, and Yan Ji from the Stony Brook University Histology Core.

## Additional information

### Funding

| Funder | Grant reference number | Author |
| --- | --- | --- |
| National Cancer Institute | RO1CA236389 | Nancy C Reich |
| Carol M Baldwin Breast Cancer Research Award | | Nancy C Reich |
| Catacosinos Cancer Research Award | | Oleksi Petrenko |

The funders had no role in study design, data collection and interpretation, or the decision to submit the work for publication.

### Author contributions

Stephen D'Amico, Conceptualization, Data curation, Investigation, Methodology, Writing – review and editing; Varvara Kirillov, Data curation, Investigation, Methodology; Oleksi Petrenko, Conceptualization, Data curation, Formal analysis, Investigation, Methodology, Writing - original draft; Nancy C Reich, Conceptualization, Resources, Formal analysis, Funding acquisition, Project administration, Writing – review and editing

### Author ORCIDs

Nancy C Reich (iD) https://orcid.org/0000-0003-4367-6097

### Ethics

All animal studies were approved by the Institutional Animal Care and Use Committee at Stony Brook University (IACUC 269685). Animals were handled according to approved IACUC protocols.

Reviewer #1 (Public review): https://doi.org/10.7554/eLife.92559.2.sa1
Reviewer #2 (Public review): https://doi.org/10.7554/eLife.92559.2.sa2
Author response https://doi.org/10.7554/eLife.92559.2.sa3

---

## Additional files

### Supplementary files

• MDAR checklist

### Data availability

Human PDAC expression profiles from The Cancer Genome Atlas (TCGA) were downloaded as z-scores from cBioPortal (http://www.cbioportal.org), along with additional tumor and clinical annotations. PDAC datasets from the Amsterdam UMC (AUMC; *Dijk et al., 2020*) and PanCuRx Translational Research Initiative (COMPASS; *Chan-Seng-Yue et al., 2020*) were used as described. The RNA-Seq data has been deposited in Dryad: https://doi.org/10.5061/dryad.1vhhmgqzc. The scRNA-seq data have been deposited in the GEO/SRA database under accession code GSE132582 (https://www.ncbi.nlm.nih.gov/geo/query/acc.cgi?acc=GSE132582). Additional information and/or reagents are available from the authors on request.

The following dataset was generated:

| Author(s) | Year | Dataset title | Dataset URL | Database and Identifier |
|---|---|---|---|---|
| Stephen NR, D'AV K, Petrenko O | 2023 | Functional antagonism between STAT3 and SMAD4 regulates EMT | https://doi.org/10.5061/dryad.1vhhmgqzc | Dryad Digital Repository, 10.5061/dryad.1vhhmgqzc |

The following previously published dataset was used:

| Author(s) | Year | Dataset title | Dataset URL | Database and Identifier |
|---|---|---|---|---|
| Ischenko I, D'Amico S, Rao M, Li J, Hayman MJ, Powers S, Petrenko O, Reich NC | 2020 | KRAS drives immune evasion in a genetic model of pancreatic cancer | https://www.ncbi.nlm.nih.gov/geo/query/acc.cgi?acc=GSE132582 | NCBI Gene Expression Omnibus, GSE132582 |

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
