## [Editor Report · eLife assessment]

This study delves into the complex role of STAT3 signaling and its interplay with TGF-beta and SMAD4 in KRAS mutant pancreatic cancer. The authors demonstrate that both the presence and absence of STAT3, relative to SMAD4, can lead to poor PDAC differentiation and that STAT3 mutations affect p53-null fibroblasts with KRASG12D and induce an EMT-like phenotype. By providing **convincing** evidence, the authors were able to derive **important** insights into KRAS mutant cancers.

---

## [Referee Report · Reviewer #1 (Public review)]

Summary:

This study presents a valuable finding on the increased activity of two well-studied signal transduction pathways - STAT-3 and TGF-Beta in a specific subtype of pancreatic cancer. Specifically, SMAD4 deficient tumors (commonly observed in pancreatic cancer) are well differentiated in the presence of STAT3. Yet surprisingly, in the presence of SMAD4 in a STAT-3 deficient pancreatic cancer, the phenotype is poorly differentiated in the background of KRASGD12D. The evidence in the animal models supporting the authors' claims is solid, although including TCGA data and/or a larger number of patients would have strengthened the study. The work will be of interest to medical biologists working on pancreatic cancer and potentially the broader field.

Strengths:

Strengths are the animal models and the lead author's expertise in STAT3 signaling.

Weaknesses:

Weaknesses are the absence of correlation between the results from the animal studies and human pancreatic cancers.

---

## [Referee Report · Reviewer #2 (Public review)]

Summary:

This manuscript explores mechanisms by which STAT3 may regulate KRAS mutant cancers.

In the first set of experiments, STAT3 GOF mutants diminished the transformation of p53-null mouse embryonic fibroblasts expressing endogenous mutant KRAS(G12D) (KP MEFs) and this was dependent on direct transcriptional activation induced by phosphorylated STAT3. It appears that this is mediated via a reduction in TGFb signaling such that knockout of either TGFBR2 or SMAD4 can phenocopy the effects of STAT3 GOF mutants in KP MEFs.

In the next part of the paper, the authors used murine pancreatic ductal adenocarcinoma (PDAC)-derived cell lines bearing endogenous KRAS(G12D) and TP53(R172H) mutations (KPC) to determine the extent to which STAT3 may regulate KRAS dependency. They determined that KRAS and STAT3 KO both induced mesenchymal-like phenotypes and that TGFBR2 and SMAD4 KO induced epithelial phenotypes. The loss of STAT3 appeared to correlate with a KRAS-independent signature, and SMAD4/TGFBR2 KO could not induce epithelial phenotypes when STAT 3 was also knocked out.

Strengths:

Overall, this is an interesting paper that highlights the complicated interactions between KRAS, STAT3, and TGF beta signaling. The authors use multiple models and attempt to link data to patient cohorts.

Weaknesses:

While correlations are strong, the study would benefit from additional cause-and-effect type experiments. It would also be beneficial to better tie together the first and second parts of the paper.

---

## [Author Response]

**Reviewer #1 (Public Review):**
[...] Weaknesses are the absence of correlation between the results from the animal studies and human pancreatic cancers.

Author response: We appreciate the reviewer’s attention to the importance of human pancreatic cancer studies. In a previous study (D’Amico et al. Genes & Development 2018 doi: 10.1101/gad.311852.118), we evaluated the expression of STAT3 in human pancreatic tissue microarrays and data from the Human Protein Atlas. Mutations in Stat3 are infrequent in human pancreatic cancers, however there is a trend of decreased STAT3 activity in poorly differentiated carcinomas.

In the current study, STAT3 and SMAD4 gene signature scores (computed from KO KPC cells) were aligned with human pancreatic ductal adenocarcinoma samples from the TCGA cohort, and statistical analyses supported the selective antagonism of STAT3 and SMAD4 (Fig 4D, Fig 4E).

The complex process of EMT is difficult to characterize rigorously in human cancers. Mouse models offer an opportunity to study the relationships between cancer phenotypes and genetic alterations.

**Reviewer #2 (Public Review):**
[...] While correlations are strong, the study would benefit from additional cause-and-effect type experiments. It would also be beneficial to better tie together the first and second parts of the paper.

Author response: We understand the Reviewer’s interest in additional experiments that could further elucidate mechanisms that drive EMT and/or KRAS dependency in relation to STAT3 and TGF-beta antagonism. We previously investigated the development of mutant KRAS knockout tumors (Ischenko et al. Nature Communications 2021 doi:10.1038/s41467-021-21736) to find loss of KRAS promotes EMT, similar to loss of STAT3. Additional experiments are underway but are outside the scope of the current study.

The first part of the paper is mechanistic and used KRAS-transformed mouse embryo fibroblasts to perform in vitro studies with foci formation. The cell-based foci formation assay has been shown to best evaluate malignant transformation and oncogenic potential. In the second part we transitioned to epithelial cells and pancreatic ductal adenocarcinomas to combine mechanistic relationships with genetic models.